# Drug Dosing Considerations in Critically Ill Patients Receiving Continuous Renal Replacement Therapy

**DOI:** 10.3390/pharmacy8010018

**Published:** 2020-02-07

**Authors:** Soo Min Jang, Sergio Infante, Amir Abdi Pour

**Affiliations:** 1School of Pharmacy, Loma Linda University, Loma Linda, CA 92350, USA; 2School of Medicine, Loma Linda University, Loma Linda, CA 92350, USA; sinfante@llu.edu (S.I.); aabdipou@llu.edu (A.A.P.)

**Keywords:** antibiotics, drug dosing, acute kidney injury, continuous renal replacement therapy, pharmacokinetics, pharmacodynamics

## Abstract

Acute kidney injury is very common in critically ill patients requiring renal replacement therapy. Despite the advancement in medicine, the mortality rate from septic shock can be as high as 60%. This manuscript describes drug-dosing considerations and challenges for clinicians. For instance, drugs’ pharmacokinetic changes (e.g., decreased protein binding and increased volume of distribution) and drug property changes in critical illness affecting solute or drug clearance during renal replacement therapy. Moreover, different types of renal replacement therapy (intermittent hemodialysis, prolonged intermittent renal replacement therapy or sustained low-efficiency dialysis, and continuous renal replacement therapy) are discussed to describe how to optimize the drug administration strategies. With updated literature, pharmacodynamic targets and empirical dosing recommendations for commonly used antibiotics in critically ill patients receiving continuous renal replacement therapy are outlined. It is vital to utilize local epidemiology and resistance patterns to select appropriate antibiotics to optimize clinical outcomes. Therapeutic drug monitoring should be used, when possible. This review should be used as a guide to develop a patient-specific antibiotic therapy plan.

## 1. Introduction

Continuous renal replacement therapy (CRRT) is commonly used in critically ill patients with acute kidney injury (AKI) due to life-threatening fluid overload and/or electrolyte abnormalities. Mortality rate from septic shock is still high (60%), despite the advancement in medicine [1]. The 2016 Sepsis Surviving Guideline suggests clinicians to administer the appropriate dose of antibiotic as soon as the septic shock is recognized [2]. Many drugs used in critically ill patients with AKI can be titrated to effect (pain medications, sedatives, and vasopressors). However, the serum concentrations of most antibiotics cannot be measured or titrated in the clinical setting. Consequently, the empiric dosing regimen must be correct with the first dose. This paper will focus on antibiotic agents and how to optimize antibiotic dosing strategies (time- vs. concentration-dependent) depending on their pharmacodynamics (PD) targets.

## 2. Drug Dosing Challenges for Clinicians 

Continuous renal replacement therapy (CRRT) has been the preferred treatment choice for hemodynamically unstable patients with acute kidney injury (AKI). It is better tolerated by critically ill patients than intermittent hemodialysis (IHD). However, delivery of CRRT at any given hospital can vary due to differences in anticoagulation (heparin vs. sodium citrate), CRRT modality (continuous veno-venous hemofiltration (CVVH), continuous veno-venous hemodialysis (CVVHD), continuous veno-venous hemodiafiltration (CVVHDF), replacement solution administration, and prescribed effluent rates. Additionally, hybrid renal replacement therapy (RRT), such as prolonged intermittent renal replacement therapy (PIRRT) or sustained low-efficiency hemodialysis (SLED), adds another layer of complexity of appropriate antibiotic dosing in critically ill patients. Clinicians are hesitant to utilize hybrid RRTs, despite the benefits of IHD and CRRT (ex., cost-effective and well-tolerated by hemodynamically unstable patients [3]). There is a scarcity of pharmacokinetic (PK) data in hybrid RRT and inconsistency of variability in hybrid RRT operating parameters. Thus, clinicians often need to extrapolate PK data from in vitro reports, case reports, or studies conducted from a variety of RRT modalities to determine drug dosing. A study in critically ill patients receiving CRRT showed that only 53% of those received ceftazidime treatment, and 0% of those receiving cefepime treatment achieved the desired pharmacodynamic targets against *P. aeruginosa* [4]. A drug like piperacillin has wide interindividual variability in critically ill patients and may not reach the probability of target attainments (PTA) [5]. Therapeutic drug monitoring (TDM) is being utilized for β-lactams (piperacillin or meropenem) in some hospitals to individualize the therapy [6]. However, it is not available at the majority of medical centers. For drugs that require TDM, ensure the blood sample is collected outside the CRRT system or from a pre-filter port to avoid underestimation of the patient’s drug concentration. A working knowledge of how CRRT is delivered to a particular patient is critical to develop an appropriate personalized pharmacotherapy [7,8]. The goal of therapy in critically ill patients with sepsis should be the administration of appropriate antibacterial drugs within one hour of sepsis recognition to decrease morbidity and mortality [9]. Each one-hour delay in effective antibiotic administration is associated with increased mortality [10]. The clinician must ensure that the “appropriate” drug is selected with the adequate dose accounting for CRRT clearance and the altered PK in critical illness.

## 3. Pharmacokinetic Changes in Critically Ill Patients with Sepsis

Due to increased capillary permeability and fluid accumulation from sepsis, volume of distribution (Vd) of antibiotic is increased. Hydrophilic antibiotics like β-lactams are more affected than lipophilic antibiotics such as fluoroquinolones. Up to one-half of all critically ill patients may develop hypoalbuminemia, which will directly affect the drug Vd by increasing the unbound drug fraction of highly protein-bound drugs [11]. An increase in unbound drug concentrations may enhance pharmacologic effects and increase the toxicity risk. Additionally, higher free drug concentrations increase the amount of drug available to be dialyzed by RRT. Critically ill patients with AKI will likely have a decreased non-renal clearance (CL_NR_) compared to healthy individuals but higher CL_NR_ compared to end-stage renal disease (ESRD). The difference between some drugs’ CL_NR_ almost doubled in critically ill patients with AKI compared to ESRD patients. For example, imipenem’s CL_NR_ was 95 mL/min in critically ill patients with AKI, whereas it was 50 mL/min in ESRD patients [12,13]. Vancomycin’s CL_NR_ was reported to be 15 mL/min in AKI patients compared to 5 mL/min in ESRD patients [14,15,16]. Surprisingly, meropenem’s CL_NR_ for patients with normal renal function (CL_NR_ 45–60 mL/min) [17,18,19] was reported to be similar to AKI patients’ (CL_NR_ 40–60 mL/min) [20,21].

## 4. Types of Renal Replacement Therapy

Drug removal is influenced by the mode of RRT, frequency of dialysis, and flow rates of RRT. Filters that are used in RRT influence the drug removal. However, high-flux filters are commonly used in current clinical settings. Thus, the influence of filter types will not be discussed in this paper. Increased RRT frequency will result in more drug removal. Since there is no data to prove a superiority of any type of RRT [22], it is important to consider how fast the antibiotic is being removed by different RRT modalities. Generally, the slowest rate between blood and effluent (dialysate and/or ultrafiltrate) rate is the one that ultimately determines solute clearance. For example, in intermittent hemodialysis, the dialysate rate is usually twice the blood flow rate; consequently, it is blood flow that determines the dialytic clearance. 

Intermittent hemodialysis provides a rapid (usually 3–5 h) RRT that is often performed thrice-weekly in outpatient regimens. IHD provides much higher extracorporeal drug clearance than other RRTs. For example, blood flow rate ranges between 250–450 mL/min for IHD, 150–400 mL/min for PIRRT, and 150–250 mL/min for CRRT. The dialysate flow rate ranges 500–800 mL/min for IHD, 100–300 mL/min for PIRRT, and 1–3 L/h for CVVHD and CVVHDF. Ultrafiltration rates are 1–3 L per 3–5 h of IHD, 1–4 L per 6–12 h of PIRRT and 1–3 L/h for CVVH and CVVHDF [23]. The main drug removal during non-IHD treatment for the patients would be CL_NR_. Many drug package inserts provide drug dosing recommendations for hemodialysis patients. However, these dose recommendations are not applicable in critically ill patients receiving IHD, since these PK data are predominantly generated in ESRD patients. PK parameters are not only markedly different in this patient population, but also, hypercatabolic critically ill patients may require more frequent IHD (>3 times weekly) to control electrolyte and waste product removal [24]. Drug dosing regimens that are appropriate for a thrice-weekly hemodialysis schedule are unlikely to benefit patients needing IHD five to seven times per week. 

Prolonged intermittent renal replacement therapy or sustained low-efficiency dialysis is a type of hybrid RRT to achieve the benefits of IHD and CRRT. The PIRRT is usually operated for 6–10 h daily. It can be used in hemodynamically unstable patients and is cost-effective compared to CRRT [3]. Moreover, it can provide an opportunity for procedures or physical therapy during downtime without limiting dialytic treatment. Yet, the inconsistency with PIRRT regimens complicates drug dosing. Prescriptions for PIRRT are different from institution to institution, and drugs that needs to be given every 6–8 h must sometimes be administered while PIRRT is operating. This leads to questions such as “Do you administer the drug before, during, or after PIRRT?” and “Do I need to give higher doses while PIRRT is running than when PIRRT is turned off?” The complexity of drug dosing in PIRRT is also reflected in the survey of 69 experienced American critical care pharmacists. They were surveyed for PIRRT dosing recommendations for cefepime, ceftaroline, daptomycin, levofloxacin, meropenem, and piperacillin/tazobactam [25]. Up to nine distinct regimens per antibiotic were recommended, and total daily doses varied from 4- to 12-fold. The wide drug dosing ranges by experienced pharmacists show that guidance with PIRRT antibiotic dosing is inadequate and varies widely, leading to lack of standardization. 

Continuous renal replacement therapy is intended to run 24 h/day, which allows hemodynamically unstable patients with AKI to utilize it. Yet, CRRT interruptions do often occur. It has been reported that the prescribed CRRT clearance was overestimated by 24% compared to the actual delivered clearance at an academic medical center [26]. This shows that prescribed CRRT doses may significantly be overestimated compared to the actual delivered clearance, leading to potential drug accumulation. Clinicians should adjust the drug dosing regimen if CRRT has been interrupted. Unlike IHD, drug clearance for CRRT can be calculated (Table 1). In order to calculate the CRRT drug clearance, clinicians need to determine the sieving coefficient (SC) and saturation coefficient (SA) for solutes. The SC and SA describe a solute’s ability to cross the hemodiafilter membrane in CVVH and CVVHD, respectively. Both terms are expressed as the concentration of drug/solute in ultrafiltrate or dialysate relative to plasma. The SC and SA can range from 0 (no drug clearance via CRRT) to 1 (drug is freely cleared by CRRT). These terms are often used interchangeably. The SC can be determined by measuring solute concentrations from ultrafiltrate and the solute concentration from the patient’s blood. It can be calculated by [concentration in ultrafiltrate]/[concentration in blood]. In the absence of drug concentrations, it can be estimated as SC = 1 − protein binding (PB). Drug PB rates in critically ill patients can be difficult to determine, because PK studies are limited; consequently, “normal” rates are often used. The latter approach is less accurate, because critically ill patients may have different PB rates compared to healthy subjects from low serum albumin concentrations and retained waste products. Table 1 shows equations to calculate CRRT solute clearance depending on CRRT modalities. Replacement fluids must be administered when convection is used to replace the ultrafiltrate produced to prevent hypovolemia. Replacement solutions can be infused into any of the ports (pre-filter or post-filter) that exist in the extracorporeal CRRT circuit. Drug removal by CRRT is affected by where a replacement solution is infused into the extracorporeal circuit. Drug concentration is diluted in the filter when the fluid is infused as a pre-filter, resulting in decreased drug removal in pre-filter CVVH. Conversely, the drug is removed very efficiently across the hemodiafilter membrane when the replacement solution is infused as a post-filter. Although pre-filter fluid replacement CRRT reduces solute clearance, it prevents having a filter clogging (when hematocrit concentration increases within the hemodiafilter, it results in a thick and viscous blood). Since a CRRT machine will stop when the filter is clogged, many medical centers use pre-filter CVVH, despite knowing that the CRRT solute/drug removal will be less efficient than post-filter CVVH [23]. A “correction” factor for pre-filter fluid replacement appears in Table 1.

## 5. Drug Administration Strategies in CRRT

### 5.1. Drug Specific Considerations 

Drugs’ molecular weight (MW) and PB affect RRT drug clearance. The larger the MW, the more difficult it is for the drug to cross the hemodiafilter membrane. For example, blood proteins are too large to be cleared by the membrane, highly protein-bound drugs will remain in the blood (not removed by CRRT). Blood, dialysate, and ultrafiltrate rates can be independently prescribed to meet solute and fluid removal goals in any RRT. Generally, the slowest rate between blood or dialysate + ultrafiltrate rate is the one that ultimately determines a solute clearance. Aforementioned blood flow is usually much higher than that of the effluent (dialysate + ultrafiltrate) rate in CRRT. Thus, the effluent rate determines the drug clearance by CRRT. The faster the effluent rate, the more drug removal will occur in patients receiving CRRT. However, intensity of CRRT effluent rates (25 mL/kg/h vs. 35 mL/kg/h) did not show any differences in clinical outcome [28]. Moreover, a recent Monte Carlo simulation study showed the intensity of effluent rates did not influence the PTA for selected β-lactam drugs [29].

### 5.2. Time-Dependent vs. Concentration-Dependent Antibiotics

In general, antibiotics are classified as: (1) concentration-dependent (e.g., aminoglycosides and fluoroquinolones); (2) time-dependent (e.g., β-lactams and carbapenems); or (3) both (e.g., vancomycin). Concentration-dependent antibiotics are most effective when drugs reach maximum concentration (Cmax) values relative to the target organism’s minimum inhibitory concentration (MIC). Time-dependent antibiotics are optimized when the time that the serum concentration is above the minimum inhibitory concentration (T ≥ MIC) is maximized. In antibiotics that require both concentration- and time-dependent PD targets, the PD target is defined as an area under the curve (AUC) for 24 h over MIC (AUC_0–24_/MIC). Table 2 lists published PD targets and empiric dosing recommendations for adult patients with AKI who are receiving any form of CRRT at different effluent flow rates (Qeff). Even though the Qeff has been defined in the table, the Qeff has been shown to insignificantly affect the PTA of different antibiotics [29]. 

### 5.3. Drug Administration Strategies

In general, there are three strategies to administer any given drugs: (1) standard infusion (administer drug for 30 min), (2) extended infusion (administer drug for 3–4 h), and (3) continuous infusion (administer drug for 24 h). For critically ill patients, some suggest having a higher free drug concentration (e.g., 4 × MIC) to maximize bacterial killing and suppress bacterial resistance [33]. For example, the T ≥ MIC goal for β-lactam agents is 40–60% [33]. However, patients who achieved concentrations above the MIC for the entire dosing interval (T ≥ MIC of 100%) with cefepime or ceftazidime had higher cure rates (82% vs. 33%) and higher bacterial eradication rates (97% vs. 44%) than patients with T ≥ MIC of 34–86% [35]. Thus, higher T ≥ MIC targets should be considered with serious infections in critically ill patients. Due to decreased renal clearance, underdosing of antibiotic therapy is prevalent mainly because of drug toxicity concerns [4,7,8]. In reality, there is a higher mortality rate in critically ill patients due to an infection compared to a mortality rate from a drug toxicity. Antibiotic dosing strategies should be designed to take advantage of the CRRT, depending on the drug’s PD targets.

The flowchart (Figure 1) illustrates which drug administration strategy would be advantageous, depending on the antibiotics’ PD targets. For example, continuous or extended infusion with the LD (standard infusion) strategies should be used with a time-dependent antibiotic to maximize T ≥ MIC in critically ill patients. As mentioned earlier, extended antibiotic infusions can maintain target steady-state serum concentrations in critically ill patients receiving CRRT [36]. However, these drug administration strategies are not yet considered as a standard practice [37], most likely due to lack of efficacy evidence in patients receiving CRRT. Despite the lack of study, this intervention should be considered in patients with severe infections, because it is a simple intervention that has been documented to improve the T ≥ MIC. Extended or continuous infusion may not be appropriate in concentration-dependent antibiotics like vancomycin. Vancomycin continuous infusions (16–35 mg/kg/day) with a loading dose of 15 mg/kg successfully reached target serum concentrations (20–30 mg/L) with CRRT [38]. However, 26% of patients had drug accumulations with serum concentrations greater than 30 mg/L on Day 3, leading to increased drug toxicity risk. Concentration-dependent antibiotics might pair better with bolus infusions, because they require high-peak concentrations to avoid drug-resistance from suboptimal antibiotic exposure [39]. Since patients receiving CRRT tend to be volume-overloaded, higher doses in short periods need to be administered to attain appropriate initial peak concentrations. Taccone and colleagues showed that patients receiving CRRT with severe sepsis required at least 25 mg/kg of amikacin as a loading dose to achieve target peak concentrations [40]. However, authors suggested to extended the dosing interval to >24 hours once the target peak serum concentrations are reached, due to a high amikacin concentration with a wide inter-variability [40]. Therapeutic drug monitoring should be utilized, if applicable, even with β-lactams to prevent underdosing and overdosing in critically ill patients receiving CRRT [8,41].

## 6. Conclusions

Critically ill patients with AKI receiving CRRT require higher antibiotic doses than patients with kidney dysfunction, including end-stage renal disease (e.g., higher CL_NR_, volume overload, and resistant organisms in the ICU). Drug dosing recommendations and drug administration strategies are intended to guide the clinicians to individualize therapy plans for their patients. It is indispensable to check local epidemiology and resistance patterns to select appropriate antibiotics to improve patient outcomes. 

## Figures and Tables

**Figure 1 pharmacy-08-00018-f001:**
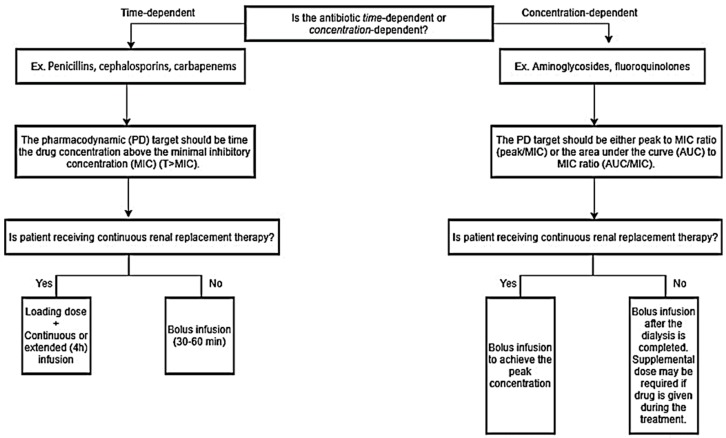
Drug administration strategy for time-dependent vs. concentration-dependent antibiotics for critically ill patients receiving renal replacement therapy.

**Table 1 pharmacy-08-00018-t001:** Equations for calculating continuous renal replacement therapy (CRRT) clearance [23,27].

CRRT Types	CRRT Clearance Equations	Solute Removal
CVVH pre-dilution	CL_CVVH(pre) =_ Q_eff_ × SC × ($\frac{\mathrm{Qb}}{\mathrm{Qb}+\mathrm{Qrep}}$)	Convection (influence of gravity)
CVVH post-dilution	CL_CVVH(post)_ = Q_eff_ × SC	Convection
CVVHD	CL_CVVHD_ = Q_d_ × SA	Diffusion (concentration gradient)
CVVHDF	CL_CVVHDF_ = (Q_u__f_ + Q_d_) × SA	Convection + diffusion
CVVH(pre) = pre-dilution continuous veno-venous hemofiltration; CVVH(post) = post-dilution continuous veno-venous hemofiltration; CVVHD = continuous veno-venous hemodialysis; CVVHDF = continuous veno-venous hemodiafiltration; SA = saturation coefficient; SC = sieving coefficient; Q_b_ = blood flow rate; Q_d_ = dialysate flow rate; Q_eff_ = effluent flow rate; Q_rep_ = replacement fluid rate; Q_uf_ = ultrafiltration rate		

**Table 2 pharmacy-08-00018-t002:** Intravenous (IV) drug dosing recommendations during CRRT [23,30,31,32,33,34].

**Medication**	Accepted PD Target	Aronoff (Q_eff_ 33 mL/min)	Hoff/Heintz CVVH (Q_eff_ 17–33 mL/min)	Jang (Q_eff_ 25 mL/kg/h)
**Aminoglycosides** (CD)				
Amikacin	Cmax/MIC ≥ 10 mg/L; AUC_0–24_/MIC ≥ 70–120 mg·h/L	7.5 mg/kg q12h or 15 mg/kg q24–72h by concentrations	10 mg/kg LD, 7.5 mg/kg q24–48h	10–15 mg/kg LD, 7.5 mg/kg re-dose when trough concentrationsn<5 mg/L
Gentamicin	Cmax/MIC ≥ 10 mg/L; AUC_0__–24_/MIC ≥ 70–120 mg·h/L	1.7 mg/kg q8h or 5–7 mg/kg q12–48h by concentrations	2–3 mg/kg LD, systemic GNR infection 1.5–2.5 mg/kg q24–48h	2–3 mg/kg LD, re-dose when trough concentrations <1 mg/L
**Penicillin** (TD)				
Piperacillin/ tazobactam	≥50% *f*T ≥ 16/4 mg/L (*P. aeruginosa*)	4.5 g q8h	3.375 g q8h 4-hour infusion	4.5 g q8h
Ampicillin/ sulbactam	≥50% *f*T ≥ 8/4 mg/L (*Acinetobacter* spp.)	N/A	3 g LD, 1.5–3 g q8–12h	3 g q8–12h
**Cephalosporins** (TD)				
Cefepime	≥70% *f*T ≥ 8 mg/L (*P. aeruginosa*)	1–2 g q12h	30minute infusion 1L/h: 1 g q8h 2–3 L/h: 1 g q6h	2 g LD, 1g q8–12h
Ceftazidime	≥70% *f*T ≥ 8 mg/L (*P. aeruginosa*)	1–2 g q12h or 2 g LD, followed by 3 g/day continuous infusion	2 g LD, 1–2 g q12h	2 g LD, 1–2 g q12h
**Carbapenems** (TD)				
Meropenem	≥40% *f*T ≥ 2 mg/L (*P. aeruginosa*)	1–2 g q12h	3-hour infusion 0.5 g q8h	1 g q8–12h
Doripenem	≥40% *f*T ≥ 2 mg/L (*P. aeruginosa*)	N/A	N/A	500 mg q8h
Imipenem	≥40% *f*T ≥ 2 mg/L (*P. aeruginosa*)	500 mg q6h	1 g LD, 500 mg q8h	500 mg q6h
Ertapenem	≥ 40% *f*T ≥ 2 mg/L (*Streptococcus pneumoniae*)	1 g q24h	N/A	1 g q24h
**Fluoroquinolones** (CD)				
Levofloxacin	Cmax/MIC 6–8; AUC_0-24_/MIC ≥ 87 mg·h/L (gram-negative); AUC_24_/MIC ≥ 50 mg·h/L (gram-positive)	500 mg q48h	500–750 mg LD, 250 mg q24h	500–750 mg LD, 250–500 mg q24h
Ciprofloxacin		400 mg q24h	200–400 mg q12–24h	400 mg q12h
**Miscellaneous**				
Colistin (CD)	Free AUC/MIC 10 mg·h/L	N/A	2.5 mg/kg q48h	5–10 mg/kg LD, 2.5–5 mg/kg q24h
Aztreonam (TD)	≥50% *f*T ≥ 8 mg/L (*P. aeruginosa*)	1 g q12h	2 g LD, 1–2 g q12h	2 g LD, 1–2 g q8–12h
Linezolid (CD/TD)	AUC/MIC = 80 mg·h/L	600 mg q12h	600 mg q12h	600 mg q12h
Vancomycin (CD/TD)	AUC/MIC = 400 mg·h/L (*Staphylococcus aureus* and *S. pneumoniae*)	1 g q24–96h	20–25 mg/kg LD, 500–750 mg q12h with TDM adjustments	25 mg/kg LD, 15 mg/kg q24h re-dose when through concentrations <15 mg/L
Daptomycin (CD/TD)	AUC/MIC of 75–237 mg·h/L for *S. pneumoniae*, 388–537 mg·h/L for *S. aureus*, 0.94–1.67 mg·h/L for *Enterococcus faecium*	8 mg/kg q48h	6–8mg/kg q24h	6–8 mg/kg q24h
CD = concentration-dependent; *f*T = free serum concentration; GNR = Gram-negative rod; IV = intravenous; LD = loading dose; N/A = not available; PO = orally; q = every; TD = time-dependent; TDM = therapeutic drug monitoring; PD = pharmacodynamics; MIC = minimum inhibitory concentration; AUC = area under the curve.				

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
