# Peer review of "Drug Dosing Considerations in Critically Ill Patients Receiving Continuous Renal Replacement Therapy"

_pharmacy, 2020, doi:10.3390/pharmacy8010018_

Round 1

Reviewer 1 Report

I have reviewed the manuscript entitled “Approach to time-dependent and concentration-dependent antibiotic dosing in critically ill patients receiving continuous renal replacement therapy.” The manuscript reviews the current challenge to optimize antibiotic dosing in critically ill patients on renal replacement therapy and attempts to provide guidance for clinicians’ dosing decision. However, overall, the manuscript lacks of the novelty and most of the text except “drug administration strategies” are repetitive from existing literature/review articles. The authors made several recommendations/suggestion without sufficient evidence, thus the manuscript seems to be more an opinion than a review article. Additionally, English language editing issues is concerning and requires extensive editing/rewriting.

This manuscript reviews the optimal dosing strategies of time-dependent and concentration-dependent in in critically ill patients receiving RRT. The majority of the text appears repetitive from the existing literature lacking novelty. Authors may emphasize the last section of dosing strategy to align with the title and to strengthen the novelty. Additionally, I have found the following issues to issues to address:

Title : time-dependent and concentration-dependent antibiotic dosing is discussed in the last section (2 paragraphs). Authors may need to change the title to reflect the overall content more accurately or provide more discussion on the topic.

Lines 49-51. “…clinicians are hesitant to utilize it due to scarcity of drug dosing data and inconsistency with utilize it due to scarcity of drug dosing data and wide variability in hybrid RRT operating parameters.” The sentence does not read well and should be rewritten to correct the English/grammar error.

Lines 53-54. “Evidence suggests that existing published CRRT drug dosing guidelines either rarely meet the PD targets for some antibiotics.” This is another English language error, and needs to be rewritten.

Line 68. “lipophilic antibiotics such as aminoglycosides..” Aminoglycosides are hydrophilic agents.

Line 76. “meropenem (60 ml/min compared to …)” Change “compared to” to “vs.” for consistency.

Line 90. “supranormal” What does this word imply? Do you mean “much higher than other RRT”? IHD cannot result in higher drug clearance than drug clearance with normal kidney function. Please explain and provide reference for this statement if any.

Lines 154 “usually many times that of the effluent.” Consider changing “many times” to “much higher”

Line 155. “the most determinants of drug clearance…” The sentence does not read well and not clear.

Figure 1. It is not clear if this algorithm is for patients with CRRT vs. other RRT OR CRRT vs. no RRT. For example, for patients with NO CRRT receiving concentration-dependent agents, authors recommend bolus infusion “after the dialysis is completed” implying CRRT vs. other RRT. However, for patients with no CRRT receiving time-dependent agents, authors recommend bolus infusion. If a bolus is given during IHD/PIRRT, drug concentration may not attain the PD target adequately. Please clarify and modify if necessary. Additionally, carbapenems and aztreonam are included in beta-lactams.

Thank you for the opportunity to review your manuscript.

Author Response

Thank you for taking time to review the paper. Your concern is understandable. You will find English has been edited for the current manuscript. As you have mentioned, several authors have already written review articles on this topic. Moreover, there is another review article within the current issue on drug dosing on slow less-efficiency dialysis. It covers an individual antibiotic in critically ill patients that I did not feel the need to include in my manuscript. I have tried to bring a novelty by creating the figure for the drug administration strategy flowchart to guide clinicians. I also changed this article to an opinion article instead of a review article.

Title : time-dependent and concentration-dependent antibiotic dosing is discussed in the last section (2 paragraphs). Authors may need to change the title to reflect the overall content more accurately or provide more discussion on the topic.

The title has been changed.

Lines 49-51. “…clinicians are hesitant to utilize it due to scarcity of drug dosing data and inconsistency with utilize it due to scarcity of drug dosing data and wide variability in hybrid RRT operating parameters.” The sentence does not read well and should be rewritten to correct the English/grammar error.

This sentence has been rewritten.

Lines 53-54. “Evidence suggests that existing published CRRT drug dosing guidelines either rarely meet the PD targets for some antibiotics.” This is another English language error, and needs to be rewritten.

This sentence has been rewritten.

Line 68. “lipophilic antibiotics such as aminoglycosides..” Aminoglycosides are hydrophilic agents.

Thank you for catching this error. AMG has been changed to fluoroquinolones.

Line 76. “meropenem (60 ml/min compared to …)” Change “compared to” to “vs.” for consistency.

The change has been made.

 Line 90. “supranormal” What does this word imply? Do you mean “much higher than other RRT”? IHD cannot result in higher drug clearance than drug clearance with normal kidney function. Please explain and provide reference for this statement if any.

This sentence has been revised.

Lines 154 “usually many times that of the effluent.” Consider changing “many times” to “much higher”

The change has been made.

Line 155. “the most determinants of drug clearance…” The sentence does not read well and not clear.

This sentence has been revised.

Figure 1. It is not clear if this algorithm is for patients with CRRT vs. other RRT OR CRRT vs. no RRT. For example, for patients with NO CRRT receiving concentration-dependent agents, authors recommend bolus infusion “after the dialysis is completed” implying CRRT vs. other RRT. However, for patients with no CRRT receiving time-dependent agents, authors recommend bolus infusion. If a bolus is given during IHD/PIRRT, drug concentration may not attain the PD target adequately. Please clarify and modify if necessary. Additionally, carbapenems and aztreonam are included in beta-lactams.

Thank you for bringing a valid concern. The title of the figure has been modified to address your concern. Instead of "beta-lactams", penicillin and cephalosporin have been added in the box.

Reviewer 2 Report

Review of the manuscript ID: pharmacy-696743
Type of manuscript: Review ,

Submitted to section: Clinical Pharmacy,

Title: Approach to time-dependent and concentration-dependent antibiotic  dosing in critically ill patients receiving continuous renal replacement therapy  

The manuscript entitled “Approach to time-dependent and concentration-dependent antibiotic  dosing in critically ill patients receiving continuous renal replacement therapy „ is a description of dialysis methods used in critically ill patients with acute kidney injury and present recommendations for commonly used antibiotics in these cases. Although in the literature review for the subject "antibiotics and continuous renal replacement therapy" we can find as many as over 2,600 references, only 30 were selected for the article to briefly describe the comprehensive topic. After minor correction this review may be a good guide for the clinical pharmacist or practitioner starting for practicing in this specific field.

Minor neglects in the manuscript that need to be corrected before publishing is marked in pdf in the form of comments (attached pdf).

Author Response

Thank you for your insight. I have responded to each comment in the pdf file that you have provided. There is another review article within the current issue on drug dosing on slow less-efficiency dialysis. As a reviewer for that manuscript, I am aware that antibiotic agents are individually discussed. Thus, I did not feel the need to include details in my manuscript. To reflect your concern, I have changed the type of article to an opinion paper.
